# Hygienic Behavior of *Apis mellifera* and Its Relationship with *Varroa destructor* Infestation and Honey Production in the Central Highlands of Ecuador

**DOI:** 10.3390/insects12110966

**Published:** 2021-10-25

**Authors:** Diego Masaquiza, Junior Vargas, Nelsón Ortíz, Rodrigo Salazar, Lino Curbelo, Anisley Pérez, Amilcar Arenal

**Affiliations:** 1Sede Orellana, Escuela Superior Politécnica de Chimborazo, El Coca 220150, Ecuador; junior.vargas@espoch.edu.ec (J.V.); nelson.ortiz@espoch.edu.ec (N.O.); rodrigo.salazar@espoch.edu.ec (R.S.); 2Centro de Estudios para el Desarrollo de la Producción Animal, Universidad de Camagüey “Ignacio Agramonte Loynaz”, Camagüey 74650, Camagüey, Cuba; lino.curbelo@reduc.edu.cu; 3Facultad de Ciencias Agropecuarias, Fructuoso Rodríguez Pérez, Universidad de la Habana, San José de las Lajas 32700, Mayabeque, Cuba; anisley.perez@unah.edu.cu; 4Centro de Biología Molecular, Universidad de Camagüey “Ignacio Agramonte Loynaz”, Camagüey 74650, Camagüey, Cuba; amilcar.arenal@reduc.edu.cu

**Keywords:** hygienic behavior, *Apis mellifera*, *Varroa destructor*, infestation rates, production, honey

## Abstract

**Simple Summary:**

The honey bee (*Apis mellifera*) is an insect that has a relevant role in natural and agricultural ecosystems due to its leading role in the pollination of crops that are part of humanity’s food chain. Even in the face of the modernization and the intensification of agriculture, the honey bee has maintained its economic importance due to the value generated by its products. At present, when attempting to improve the characteristics of bees, it is important to evaluate variables such as hygienic behavior, *Varroa* infestation rates, and honey production as a basis for improvement plans in search of increasing productive yields at altitudes 2600 m above sea level (m.a.s.l.). The strength of bees against parasites, and therefore the better development of their colonies, was determined, resulting in a healthy colony with an increase in honey production. The altitude and the hygienic behavior of bees in the central highlands showed an inverse relationship. There was no relationship between infestation rates and production; it is proposed that environmental factors do not modulate *Varroa* levels or honey production.

**Abstract:**

The aim of this research was to analyze the relationship among hygienic behavior (HB), *Varroa destructor* infestation, and honey production in the central highlands of Ecuador. Overall, 75 honey bee colonies were evaluated before, during, and after production at three altitude levels (2600–2800, 2801–3000, and >3000 m.a.s.l.). The hygienic behavior percentage of the colonies was determined by the pin-killing method, and the colonies were classified into three groups: high HB (>85%), mid HB (60.1–85%), and low HB (≤60%). *Varroa* infestation was diagnosed as well, and honey production was evaluated only during production. HB was high and heterogeneous, averaging 80% ± 9.7%. Its highest expression was observed at lower altitudes. The infestation degree was low (3.47% ± 1.56%), although the mite was detected in all colonies upon sampling. A negative correlation was observed between HB and *Varroa* infestation in the first sampling (−0.49 **), suggesting that the high- and mid-altitude HB colonies underwent the lowest infestation rates, regardless of sampling. The correlations between HB and production were significant (0.26 *), indicating a positive effect of HB on production, meaning that colonies with high HB obtained the highest honey production (25.08 ± 4.82 kg/hive). The HB of bees showed an inverse relationship with altitude and it tended to reduce the effect of *Varroa* infestation, favoring honey production and, thus, suggesting the feasibility of selecting colonies with high HB.

## 1. Introduction

Today, honey bees are threatened by multiple factors such as the application of crop pesticides, fragmentation and loss of habitats, and the presence of pathogens and parasites [1,2,3]. Lately, the most disturbing factor is the presence of the mite *Varroa destructor*, which is the main threat to apiculture in the region [4]. This problem is also present worldwide [5,6,7]. This parasite drastically reduces the production of honey and other bee products [8].

Vaziritabar et al. [9] indicated that environmental conditions affect mite population development. However, it is more likely that this is observed through the indirect effect of environmental factors that regulate the numbers of bee brood or the activity of certain host defense behaviors [10].

Internationally, chemical control is the most widely used method. However, it can lead to the development of acaricide-resistant mites, raise production costs, cause toxic effects on bees and man, and contaminating hive products, making it difficult for its commercialization [11]. At present, other forms of the fight against the parasite are being developed based on the ability of bees to develop their defense mechanisms for survival.

Among them is hygienic behavior [12], wich in several studies was found to allow maintaining infestation rates at viable levels with the development and production of colonies [13]. High hygiene values in apiaries lie in the health and economic importance that this behavior represents for the colonies. This translates into healthier, more productive colonies, with greater pollinating action on crops [14]. Some populations of *A. mellifera* show mechanisms that allow these bee populations to coexist with the mite for longer periods without requiring any acaricidal treatment in the hive [15]. The hygienic behavior (HB) in the honey bee (*A. mellifera*) is the ability that workers have to detect [16], uncap, and remove diseased offspring (dead or parasitized) from inside the cells of a honeycomb from the brood chamber to the outside of the colony [9,17].

The mechanisms used for the breeding selection programs are the HB, the low attractiveness of the brood, the suppression of the reproduction of the mite, and the hygienic sensitivity to varroa [18]. HB is a heritable genetic trait and high enough (>0.5) to be taken into account in *A. mellifera* breeding programs to improve the vitality of the strains [19]. HB is evaluated by several methods, including removing the offspring infested with *V. destructor* [9]. Freezing with liquid nitrogen a section of the comb with the capped brood [20], and using the sacrifice of the pupae by puncture with a needle or pin [21,22]. The latter is recommended in Europe as a standard in selection programs since it shows a positive correlation with the elimination of varroa-infested pupae [15].

The importance of the hygienic behavior of honey bee colonies in association with parasite control and the bacterial and mycotic diseases of the brood is well known [23]. The mite (*Varroa destructor*) infested adult honey bees are found with malformed or flawed and stunted with deformed wings. The bees will uncap and cannibalize the pupae, which indicates progressed mite damage of chewed down brood [24]. The parasite destroys the mechanical protective barriers of the integument and impairs the immune system of the bees. Paray and Gupta, in 2017 [25], indicated that the benefit-cost ratio decreases with an increase in the level of *Varroa* mite infestation. However, the relationship of hygienic behavior with honey production remains unclear.

Studies of bee infestation by Varroa at altitudes above 2800 m.a.s.l. are scarce. Therefore, the impact of the parasite on the bee above 2800 m.a.s.l. and the interaction with bees is unknown. In that sense, this paper aimed to evaluate the hygienic behavior of honey bees (*Apis mellifera*) and its relationship with *Varroa destructor* infestation and honey production at different altitudes in the Ecuadoran highlands.

## 2. Materials and Methods

This research was conducted in 2017. The territory presents the particularity of being traversed from north to south by the mountainous system of the Andes. The climate of the center area of Ecuador classifies as a temperate semi-wet to humid. It is warm and dry in the valleys and high cold mountain on the paramos, over 3400 m above sea level. The temperature is linked to height (i.e., between 1500 and 3000 m.a.s.l.). The average values vary between 8 and 20 °C, with a temperature gradient of less than 5 °C for every 1000 m high. The altitude also influences the amount of rain that precipitates because the cold air has little capacity to retain moisture so that few rainfalls occur. However, there are two defined stations: wet or winter (October to May) and dry or summer (June to September). The average rainfall varies between 800 and 1500 mm/year [26]

Overall, 15 apiaries (75 colonies) were studied in the provinces of Tungurahua and Chimborazo (Table 1). Samples were collected in March–April (before honey production), May–July (during production), and August–September (after production).

### 2.1. Criteria for Inclusion and Exclusion of Apiaries and Hives

According to the characterization of beekeepers [27], inclusion and exclusion criteria were considered to locate apiaries that met the requirements to enter the investigation.

Inclusion criteria were as follows:
Apiaries with Langstroth hives;Good strength of the selected colonies (seven combs covered with bees that contained an average of three breeding combs each, which is considered good strength according to [9]);Honey production per hive above the national average (10.2 kg) [28];No application of varroa treatment before the study;No introduction of queens in recent years.


Exclusion criteria were as follows:
Swarm hives (exploration every 15 days);Transhumance of the apiary;Refusal of the beekeeper to participate in the study.


In this case, from eighteen apiaries at the study beginning, three were excluded: two from the province of Tungurahua (for swarming and transhumance) and one from Chimborazo (refusal of the beekeeper). The hives under study had a breeding chamber and two half honey supers. In addition, work was carried out during the same period (March–September), and the hives under evaluation were the same in all three samples and for all experiments.

### 2.2. Sample Collection, Analysis, and Evaluation

The hygienic behavior, infestation rate of *Varroa*, and yields of all 75 colonies (six apiaries from Tungurahua and nine from Chimborazo) were determined. The methodology used in each case is presented below.

#### 2.2.1. Hygienic Behavior (HB)

The evaluation was made in each colony; we chose two brooded combs containing sixteen to seventeen-day-old pupae (pink-eye pupae). We selected a ten × ten cell region, and the pupae were pin-killed. The comb was returned to the colony for evaluation after 24 h [22]. The total hygienic behavior (THB) formula was recorded using the formula [29].
$$\mathrm{THB}=\frac{\text{Number of pupae removed}}{\text{total number of pierced cells}}\times 100$$


The three HB evaluations were averaged and classified [30] with slight modifications. Colonies uncapped and removed with more than 85% of sacrificed breeding were classified as high HB, while those removed with 60.1% to 85% of breeding as mid, and those removed with less than or equal to 60% of breeding as low.

#### 2.2.2. Infestation Rates (IR)

A total of 150–200 honey bees were removed from the center of the brooding chamber and placed in a container with water and commercial detergent [31]. The mites detached from the honey bee bodies were placed on white trays and quantified. The infestation rate was determined by the formula:
$$\mathrm{IR}=\frac{\text{Number of varroas mites}}{\text{Number of bees}}\times 100$$


#### 2.2.3. Honey Production

Each colony was weighed before and after harvest. The weight difference was considered honey yield [32]. The honey collected throughout the season was considered total honey production. The honey stored in the brooding chambers was not included.

### 2.3. Statistical Analyses

SPSS 21 was used for statistical analysis, and Kolmogorov normality tests were performed. Data are expressed as mean ± SEM. HB data were transformed using arcsin sqr (THB/100) to meet normality. Bivariate correlations (Pearson) were as follows: THB between samples, THB and altitude level, THB and IR, and THB and production. One-way ANOVA was performed and followed by a comparison of means (Bonferroni). Bivariate correlations (Spearman) were performed for the data without normal distribution for the IR as a function of altitude level, IR as a function of production, and production as a function of altitude level. Nonparametric tests were applied for two independent samples (Mann Whitney) to compare HB, IR, and honey production.

## 3. Results

### 3.1. Hygienic Behavior

The evaluation of mean hygienic behavior in apiaries revealed it to be 80 ± 9.7%, while sampling results indicated mean values of 76.31%, 83.81%, and 79% for the first, second, and third samplings, respectively. According to these criteria, the HB of all the colonies in the study could be classified as medium.

Altitude influenced the differences in the observed amount of hygienic behavior, whereby correlating the THB with different heights showed a negative result (*r* = −0.25 *; *p* < 0.05) with the THAB of the third sampling. The results of the correlation between samples may indicate that the evaluation of the HB in the before-production stage (March–April) can be used as indicative of the HB colonies during the year in the region. Higher hygiene-behavior values were present at the lowest altitude, with the difference having *p* < 0.05 (Table 2; Appendix A).

Moreover, positive correlations were observed between the hygiene-behavior percentages in the first and second samplings (*r* = 0.34 **), as well as between the middle and third samplings (*r* = 0.54 **), among the hygiene-behavior percentages (*p* < 0.005) of all three samplings.

In the second sampling, the highest hygiene-behavior percentages were observed at the first tier, showing differences (*p* < 0.05) with the second and third tiers. In the third sampling, the colonies from the third tier showed a lower HB compared to the first tier (Figure 1; Appendix A).

This difference may be attributed, among factors, to the different stages undergone by the colonies during the samplings (before, during, and after production), as well as environmental changes taking place throughout the year.

### 3.2. Evaluation of Infestation Rates

In the investigation, it was determined that varroasis was present throughout the study area. We observed an increase in colonies infested by *Varroa* with the progress of the honey production moment (90.7% before, 94.7% during, and 100% at the end). Accordingly, we must also consider changes in the behavior of the parasite throughout the year, as determined by environmental conditions.

The mean infestation rate of the three samples was 3.47 ± 1.56%, with a maximum value of 12%. During these evaluation periods, variability was present in the IR of 3.5%, 2.6%, and 4.3% for the first, second, and third samplings, respectively, showing the lowest IR during the production stage. However, throughout the production stages, the IRs were similar throughout the study area (Figure 2; Appendix A). At the beginning of the period, a negative correlation (*r* = −0.28 *) between altitude and IR was found.

Likewise, in March and April, average rates of 3.34% (12% maximum) were observed, which might have been caused by the better state of the colonies in that period in terms of population, with a large number of drones (which are more appealing to mites), as well as due to massive births prior to production.

### 3.3. Honey Production Evaluation

A mean production of honey of 25.08 ± 4.82 kg was identified, with no relationships (*r* = 0.07) or differences (*p* = 0.576) at the different altitudes.

### 3.4. Relationship between Variables: Hygienic Behavior, Infestation Rates, and Honey Production

A negative correlation was observed between HB and the infestation rate in the first sampling (−0.49 **), contrary to the second and third samplings (*r* = 0.11 and *r* = −0.12, respectively). Nonetheless, when comparing the IR based on the classified HB, the average total IR is 2.62% in colonies with high HB, 3.6% in colonies with mid HB, and 8.22% in colonies with low HB (Figure 3; Appendix A), indicating that the colonies with a higher HB underwent lower parasitic burdens.

Differences in IR were identified in the three samples wherein colonies with high and mid HB had the lowest IR. The honey bees studied were not subjected to any mite-control method in the months preceding the samplings. However, the IRs were low (3.4%), thus suggesting a process of adaptation to *Varroa destructor* in the local honey bees. Although the HB presented negative correlations with altitude and IR for *V. destructor*, there was no relationship between the latter two.

The larger production levels (26.46 kg/colony) were detected in the colonies with the higher THB. The lower levels (23.43 kg/colony) were observed in colonies with intermediate THB, with 12.9% more honey in the former.

## 4. Discussion

In the current study, the hygienic behavior values were high (80%). In the Ecuadoran highlands, the farmers select their hives somewhat arbitrarily and there is a lack of genetic crossbreeding program. Beehives with hygienic-behavior values ranging from 80–90% can be considered high HB [33]. Colonies with high HB removed more than 95% of the perforated offspring, albeit at 48 h [30]. The hygienic-behavior percentages observed in this investigation were higher than reports in Chile (20–80%) [34] and Peru (71.75%) [35]. However, they were lower than the values reported in Cuba, where an average of 90% of removal of dead broods was identified [29]. In Mexico, colonies with values higher than 86% HB were identified [36].

In addition, the importance of HB and its relationship with the health and production of honey bee colonies is underappreciated in Ecuador. This is in contrast to other countries, such as Cuba, Mexico, and Peru, where colonies with high hygienic behavior, low infestation, and production above the mean are subjected to selection processes.

The importance of maintaining high levels of hygiene in the apiaries lies in the sanitary and economic significance of this behavior for the colonies, which is translated into healthier, more productive colonies with more pollinating action on crops. However, the variability in the expression of this trait depends on the aptitude or composition of the colonies. It might be due to the distribution of workers to different tasks [37]. Similarly, in evaluations conducted in two different years, a wide range of variations was attributed to seasonal changes [38].

The high hygiene percentages during the second sampling, coinciding with honey production, may be attributed to the abundant input of nectar and pollen. This stimulates the bees to clean the hives, related to the need for space to store these products in the colony [20]. Likewise, the abundant input of nectar during honey production stimulates the posture of the queen, requiring clean cells [39]. The last criteria supported the idea of conducting serial analyses of hygienic behavior throughout the year and determining their means to evaluate honey bee populations in any region.

The high prevalence of the mite may be associated with different causes, including inadequate colony management by farmers, transhumance, the uncontrolled exchange of queens and bee material, the presence of wild bees, and the absence of breeding programs in the region [40]. A lower prevalence of the mite, 88%, was found in Mexico [30], whereas a study in the US found 90% prevalence [41].

These infestation rates were lower than those observed in Cuba (5.36%) [42], as well as lower than the values of 7.51% and 6.07% obtained for father and mother lines, respectively [29]. Meanwhile, in Mexico, infestations reached 6.76% and 6.82% [36]. This behavior could be linked to environmental effects on mites and honey bee colonies. The presence of Africanization in the apiaries needs to be studied. Recently, a report of Ecuadoran bees demonstrated Africanization [43].

The increase in IR (average of 4.4%) during the third sampling (August and September) could be associated with factors such as production, the decrease in bee populations, and the displacement of a greater quantity of mites toward adult bees because there is a reduction in the posture of the queen in this period, due to it being the end of the production season.

Contradictory results have been reported with respect to the positive correlation of *Varroa* levels with altitude [44]. At the same time, a lack of correlation between elevation and *Varroa* levels suggests that the mite has managed to adapt to the environmental conditions of the highlands since there were IRs with slight variations at the three altitude tiers. However, a genetic component could also influence the bees since the IRs found were lower than those found in populations of European genotype and similar to those of African origin [45,46].

Factors, such as the existence of other pathogens, may promote the presence and spread of *Varroa* [47]. These may include temperature and humidity, soil use, pesticide burden, and the availability of resources [40]. Nevertheless, the infestation levels observed in this study were within a nonlethal range for the colonies [48]. However, low *Varroa* infestation may lead to the appearance of diseases following subsequent declines in the yield of honey [49].

The lack of evidence for differences in honey production at different altitudes may be attributed to the evaluation taking place during the period with the highest nectar flow in the region. The principal nectar source is eucalyptus (*Eucaliptus globulus* Labill.) [27]. In addition, it is indicated that the abundance and type of flowering constitute the main factor determining production [44]. However, factors related to colony management can significantly impact honey production.

Similar results were found in Mexico, with a mean production of 27.5 kg in the fall and 21.6 kg in the spring for colonies with high HB, and a mean yield of 21.42 and 13.45 kg, respectively, for those with low HB, at altitudes below 1400 m.a.s.l. [48]. Honey production depends on the interaction of factors such as the size of the population, the continuous work of honey bees, and the environment [50]. The influence of the locality brings variability in the expression of behavioral traits, IR, and production, which can be interpreted as the sum of all abiotic and biotic components in a given environment [51]. In addition, different genotypes may vary in the degree to which their phenotypes are affected by specific environmental conditions [52].

In Mexico and Chile, *Varroa* infestation rate and hygienic behavior lack of relationship [34,53]. The effectiveness of HB in reducing IR depends on several factors as the stage of colony development, environmental conditions [54], and parasite biology [2]. Nevertheless, a study revealed that bees are dependent on self-defense or natural resistance [55]. HB is important for determining the general tolerance and resistance of bees to pests and diseases [56].

The honey bees studied in tropical areas, such as Cuba [29], showed higher HB than the findings of this research. However, the IRs were also higher, indicating the possibility of high hygienic aptitude for cleansing but without the capacity to detect *Varroa* in brooding cells as in Africanized bees. These results suggest that colonies of Africanized-honey bee descendants are less prone to *Varroa* infestations than European honey bee colonies in a variety of scenarios. However, it is suggested that quantitative differences in colony-level hygienic performance are due to the different percentages of workers dedicated to hygienic behaviors, since the number of such bees is tripled in colonies with high HB.

A study in Mexico detected no differences in *V. destructor* infestation levels between colonies with high and low HB [30]. In Brazil, high heterogeneity was found when evaluating HB and IR in Africanized honey bees [57]. However, previous results suggest that the largely hygienic colonies were more prone to having a disperse or irregular pattern of capped or uncapped cells due to their ability to detect and remove *Varroa* [58]. Similar data were found when studying Africanized honey bees (3–4%) and bees in the United States (3.3–5.1%) [59]. In addition, a study on populations of *A. m. scutellata* in South Africa revealed that the presence of *Varroa* mites was common, despite BIR never exceeding 4% [60].

In turn, it has been reported that the European honey bee colonies in Europe, Asia, and North America have undergone massive losses, compared to honey bees from other parts of the world, which have successfully survived the pathogen [61]. In Brazil, the hybrids of Africanized honey bees have shown distinct behaviors, with some resistance and tolerance to mites [62]. These individuals have high genetic identification with their African ancestor and, thus, their genotypic qualities are different from those of European bees [63]. Some behavioral traits in bees are not learned but inherited, as in the case of HB [38].

This study support the findings related to the tolerance that HB offers to the colonies toward the parasite, due to its high heritability (*h*2 = 0.65) being transmitted to other generations even if environmental conditions are different from those of its predecessors [64], as identified at the different altitude tiers in the central highlands of Ecuador. However, previous results indicated a positive correlation between the altitude and the number of adults of *V. destructor* [44,65]. Contradictorily, a higher number of mites was determined with increasing altitude, suggesting that environmental factors, such as temperature and humidity, could modify the host’s behavior but not the mite’s, since the *V. destructor* lacks stages of free life [66].

Likewise, a high correlation (*r* = 0.73) was found in Africanized bees between the variables [67], as well as in European bees (*r* = 0.17 *) [51]. Similar results were found in colonies with high HB, which produced 23% more honey than the colonies with low HB. The differences in production may be because colonies with high HB eliminate diseases and parasites more quickly [17,36]. Thus, the harmful effects of the mite are minimal, which allows superior honey production [68].

In general, the results of this work suggest that selection is possible both for a higher HB and for higher honey production. However, this does not mean that these traits are genetically linked [23]. We detected a lack of correlation between IR and honey production; although it should be noted that mite infestation levels were low, and productive yields were acceptable. This parasite can seriously affect the production of honey when IRs are greater than 5% [69,70].

Equally, it was demonstrated that there are reductions in honey production with 1% infestation, worsening with an increase in infestation as a function of the area, the weather, and other factors involved in honey production [25]. The scientific literature provides conflicting results with those of the present investigation considering the effect of *Varroa* infestation on honey production. *Varroa destructor* infestation affects the quality of lipid, protein, and honey production [71]. In addition, the lack of a relationship between IR and honey production does not mean that *Varroa destructor* is absent in the colonies. On the other hand, thanks to their defense mechanisms, the bees can tolerate its effects due to low infestations in the colonies.

## 5. Conclusions

*Varroa destructor* infestation rates and honey production are unrelated to altitude in the central highlands of Ecuador, suggesting that highland environmental conditions do not modulate *Varroa* levels or production.

The hygienic behavior of bees in the central highlands of Ecuador shows an inverse relationship with altitude. It reduces the effect of *Varroa* infestation, favoring honey production and suggesting the feasibility of selecting colonies with high HB.

## Figures and Tables

**Figure 1 insects-12-00966-f001:**
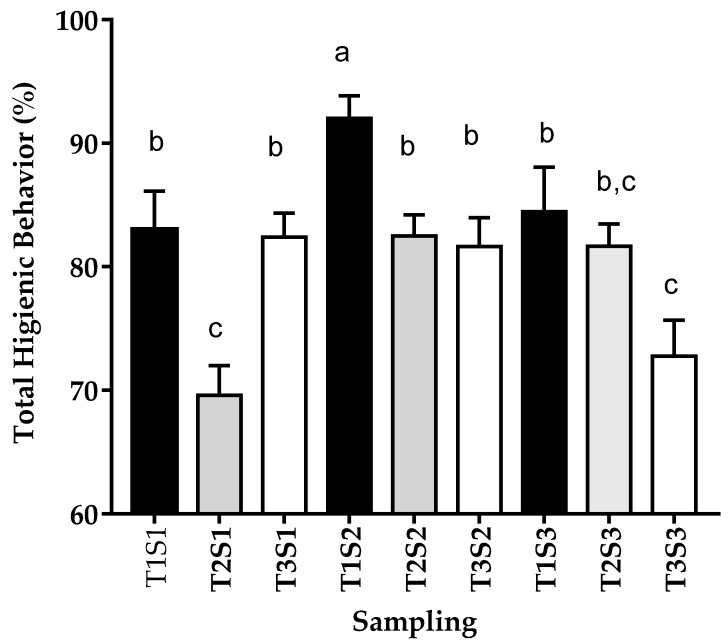
Hygienic behavior in each period of evaluation of colonies as a function of the different altitude levels of the central highlands of Ecuador. Samples were collected in March–April (before honey production, S1), May–July (during production, S2), and August–September (after production, S3). The data were divided into three altitude tiers 2600–2800 [T1], 2801–3000 [T2] and higher than 3000 [T3] meters above sea level (m.a.s.l). Data represent means ± SEM (hives/sampling: T1 *n* = 15, T2 *n* = 35, T3 *n* = 25). Different letters denote significant differences between the groups (Bonferroni).

**Figure 2 insects-12-00966-f002:**
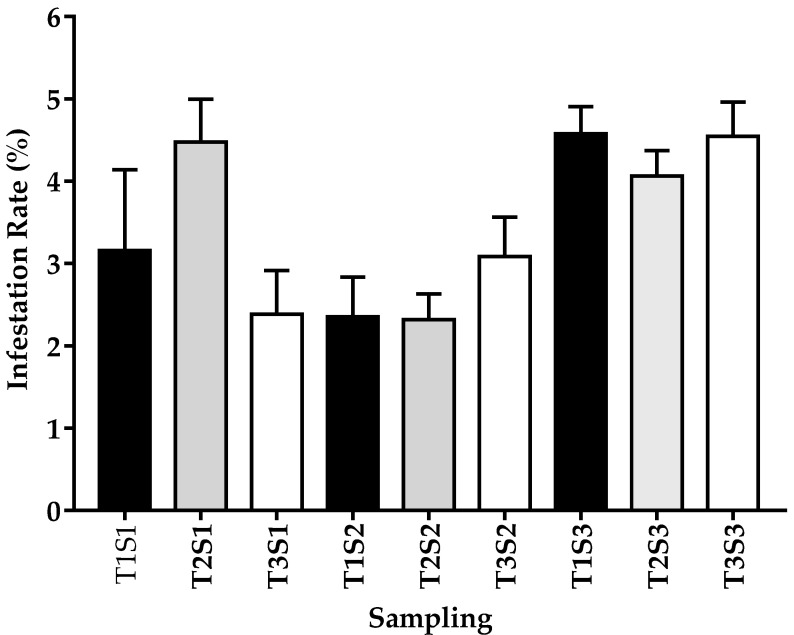
Infestation by *Varroa destructor* in each evaluation period and at different altitude tiers in the central highlands of Ecuador. Samples were collected in March–April (before honey production, S1), May–July (during production, S2), and August–September (after production, S3). The data was divided into three altitude tiers 2600–2800 [T1], 2801–3000 [T2] and higher than 3000 [T3] meters above sea level (m.a.s.l). Data represent means ±SEM (hives/sampling: T1 *n* = 15, T2 *n* = 35, T3 *n* = 25). ANOVA analysis indicated lack of significant differences between groups.

**Figure 3 insects-12-00966-f003:**
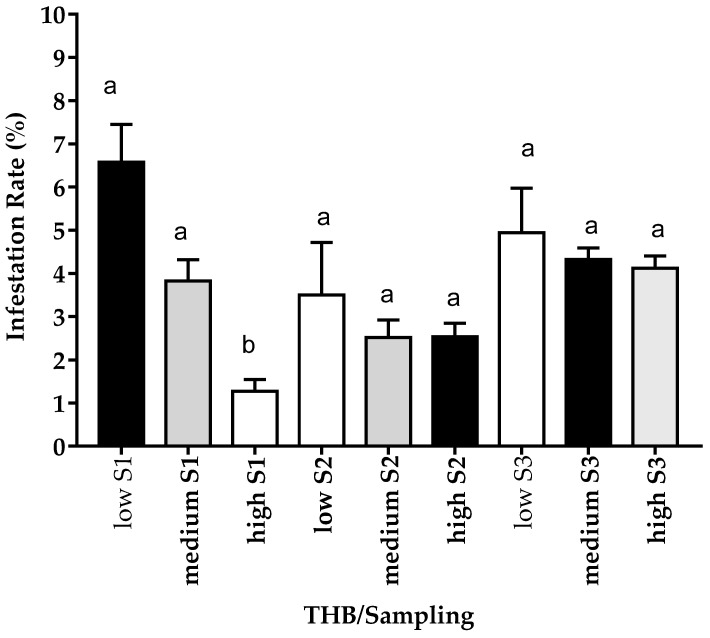
*Varroa destructor* infestation in relation to hygienic behavior (THB) (low [<=60, *n* = 11], medium [60.1–85, *n* = 43], or high [>85, *n* = 21]), in each evaluation period in the central highlands of Ecuador. Samples were collected in March–April (before honey production, S1), May–July (during production, S2), and August–September (after production, S3). Data represent means ± SEM. Different letters denote significant differences between groups (Student–Newman–Keuls).

**Table 1 insects-12-00966-t001:** Location of the studied apiaries in the Ecuadoran highlands.

Apiary	Location	Latitude	Longitude	Height (m.a.s.l.)
A1	Tungurahua	1°16′06′′ S	78°34′50′′ W	2607
A2	Tungurahua	1°22′09′′ S	78°36′19′′ W	2879
A3	Tungurahua	1°18′16′′ S	78°39′16′′ W	2936
A4	Tungurahua	1°19′02′′ S	78°39′16′′ W	3047
A5	Tungurahua	1°35′17.37′′ S	78°46′05.25′′ W	3279
A6	Tungurahua	1°33′11.2′′ S	78°42′32.4′′ W	3168
A7	Chimborazo	1°41′45.57′′ S	78°45′16.46′′ W	2939
A8	Chimborazo	1°39′26.17′′ S	78°34′38.49′′ W	2727
A9	Chimborazo	1°42′46.63′′ S	78°39′50.33′′ W	2967
A10	Chimborazo	1°35′11′′ S	78°45′09′′ W	3205
A11	Chimborazo	1°35′18′′ S	78°46′03′′ W	3262
A12	Chimborazo	1°41′34′′ S	78°40′11′′ W	2834
A13	Chimborazo	1°35′46.75′′ S	78°39′51.45′′ W	2870
A14	Chimborazo	1°43′46.5′′ S	78°36′47.6′′ W	2616
A15	Chimborazo	1°46′40.91′′ S	78°35′10.99′′ W	2863

m.a.s.l. meters above sea level.

**Table 2 insects-12-00966-t002:** Evaluation of the average hygienic behavior of *Apis mellifera* at different altitude levels of the central highlands of Ecuador.

Total Hygienic Behavior (THB %)	Altitude Levels (m.a.s.l.)		
	2600–2800	2801–3000	>3000
*N*	15	35	25
Mean	86.5 ^a^	77.9 ^b^	78.9 ^b^
SEM	10.29	8.38	8.34
Minimum	55.3	51.3	59.7
Maximum	96.7	90.7	93

Different letters denote significant differences between altitude levels. m.a.s.l. meters above sea level.

## Data Availability

The data presented in this study are available in Appendix A.

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
