# Peer review of "Hygienic Behavior of Apis mellifera and Its Relationship with Varroa destructor Infestation and Honey Production in the Central Highlands of Ecuador"

_insects, 2021, doi:10.3390/insects12110966_

Round 1

Reviewer 1 Report

The paper presented to me describes some interesting results of a field-testing study on honey bees and Varroa infestation of their colonies. Though I found the paper interesting it does need in-depth editing.

First of all, the English language should be corrected. It is difficult to read and understand, what the authors actually had in mind. Especially, the Authors should pay attention to use proper terminology.

Eg. Hive is not apiary. Hive is the box in which a bee family lives. So they tested 15 apiaries and 75 colonies/families which were living in 75 hives. We rather say bee products, not bee-colony-made products. We do not call young larvae pups, I would suggest offspring, or simply bee larvae or pupae. In a few instances the word breeding is used instead of brood.

The Introduction is quite short and does not explain why the authors found it interesting to test apiaries located on various altitudes. I also miss a bit more outlook on the differences in hygenic behaviour among bee races or breeding lines. Based on such outlook the Authors should add their expectations and next discuss the achieved result based on their expectations and already published studies.

The presentation of the result should by also modified. Fig. 1 the colums would be more infromative if grouped for altitudes instead of seasons: so T1S1 T1S2 T1S3 .... You can also use Low/medium/high instead of t1 T2 T3 and spring / summer /fall instead of S1 S2 S3.

Tab. 3. Why show the results for the provinces? If it is important please state in the Methods what is the difference between provinces. A map with the location of apiaries would be helpful.

Fig. 3 please use low / medium / high instead of <60, 60 - 80 > 80.

Other corrections:

When first using "HB", please clarify for what it stands for.

When describing the work of Paray and Gupta please add the year of publication and add to the bibliography.

Author Response

First of all, we greatly appreciate the reviewer’s comments.

First of all, the English language should be corrected. It is difficult to read and understand, what the authors actually had in mind. Especially, the Authors should pay attention to use proper terminology.

Eg. Hive is not apiary. Hive is the box in which a bee family lives. So they tested 15 apiaries and 75 colonies/families which were living in 75 hives. We rather say bee products, not bee-colony-made products. We do not call young larvae pups, I would suggest offspring, or simply bee larvae or pupae. In a few instances the word breeding is used instead of brood.

We reviewed carefully, and referee three made suggestion to improve english

The Introduction is quite short and does not explain why the authors found it interesting to test apiaries located on various altitudes. I also miss a bit more outlook on the differences in hygenic behaviour among bee races or breeding lines. Based on such outlook the Authors should add their expectations and next discuss the achieved result based on their expectations and already published studies.

In the new version, we eliminated the focus of the bee races or breeding lines. The manuscript lack evidence to get in deep into these elements. We hope that the new introduction gives a better presentation of the manuscript line.

The presentation of the result should by also modified. Fig. 1 the colums would be more infromative if grouped for altitudes instead of seasons: so T1S1 T1S2 T1S3 .... You can also use Low/medium/high instead of t1 T2 T3 and spring / summer /fall instead of S1 S2 S3.

We think it is better to keep as we described in the figure and through the text. T represents the three altitude tiers 2600-2800 [T1], 2801-3000 [T2] and higher than 3000 [T3] meters above sea level (m.a.s.l). we believe that Low/medium/high is too general and difficult to represent what we want to.

Tab. 3. Why show the results for the provinces? If it is important please state in the Methods what is the difference between provinces. A map with the location of apiaries would be helpful.

We decided to eliminate the province report. Both provinces are neighbors, and the sampling was designed without considering the province but the altitude tiers.

Fig. 3 please use low / medium / high instead of <60, 60 - 80 > 80.

Done!!!

Other corrections:

When first using "HB", please clarify for what it stands for.

Done!!!

When describing the work of Paray and Gupta please add the year of publication and add to the bibliography.

Done!!

Reviewer 2 Report

This study reports hygienic behavior (in pin-killed test), Varroa infestation, and their impact on honey production in high altitude region of  Ecuador. In total, 75 strong  bee colonies from three altitude zones  (2.6-2.8; 2.8-3.0, and >3.0km above sea level), all showing moderate Varroa infestation  were investigated.  This study demonstrated statistically significant negative correlation between hygienic behavior and Varroa infestation, and positive correlation between  hygienic behavior and  honey production. The MS is well written, results are original and will be of interest to world bee science community. I have only few suggestions.

Page 2, Materials and methods ( or discussion ). Not all readers are familiar with Ecuador highlands  climate and high altitude beekeeping. Add information  on the average temperatures in these areas, and if the bees in these locations have broodless period. Broodless periods during winter are known to result in decreased numbesr of Varroa mite. 

Table 1.
In addition to Table 1,  locations of the apiaries and altitudes should be shown on a map. This would make easier to see relative location and the distances between them.

Page 3. Section 2.1.  Specify if the  bees were Africaniszed or not.  Are these high altitude bee families/queens  locally produced or imported  from low altitude areas. 

Author Response

First of all, we greatly appreciate the reviewer’s comments.

Page 2, Materials and methods ( or discussion ). Not all readers are familiar with Ecuador highlands  climate and high altitude beekeeping. Add information  on the average temperatures in these areas, and if the bees in these locations have broodless period. Broodless periods during winter are known to result in decreased numbesr of Varroa mite. 

We have added a paragraph with the Ecuadorian altiplano.

Table 1.
In addition to Table 1,  locations of the apiaries and altitudes should be shown on a map. This would make easier to see relative location and the distances between them.

The authors believe that the data given in the MS is enough to test our hypothesis and to bring the readers the idea of altitude and honey production.

Page 3. Section 2.1.  Specify if the  bees were Africaniszed or not.  Are these high altitude bee families/queens locally produced or imported  from low altitude areas. 

They are produced locally. We don’t have evidences to confirm the Africanization of the bees. We are doing a parallel job to study the Africanization in the altiplano.

Reviewer 3 Report

This article is very interesting especially since Varroa destructor is a mite that creates damage to beekeeping. I believe the article can be accepted following some minor revisions:

-page 1 simple summary- line 6, I suggest to change "infestation rates against Varroa" in “Varroa infestation rates”

-page 1 abstract- lines 5-6 the percentages reported are not those considered in the materials and methods. please check.

-page 2 introduction- line 18 (>0.5) unit of measurement is missing. Please add it.

-page 2 introduction-lines 27-28, 29-31, 31-32 bibliographic references are missing. please add them.

-page 3 materials and methods- 2.2.1 section- why is only the value of the uncapped brooded cells divided by the brood? is it a number that comes out of a division or is it just a way to indicate a unit value? written like that it is not clear.

-page 4 materials and methods- 2.2.1 section- lines 1-4 the percentages reported are not those considered in the abstract. please check.

-page 4 materials and methods- 2.2.2 section. In the scope of the work the authors talk about Varroa infestation which is the parameter they went to evaluate. Why is it called "bee infestation rates (BIR)"?

-page 4 materials and methods- 2.3 section. What tests for normality were applied?

-page 5 and page 5 figure 1 and 2 Considering that the data are grouped by sampling and altitude they are many so I suggest to apply a correction (example Bonferroni) during statistical analysis.

-page 6 lines 8-16. “The increase in BIR (average of [….] massive births prior to production.” This whole part should be in the discussion. Please move it.

-page 7 3.4 section- In the text the abbreviation THB is used but it does not seem to correspond to the formula that was provided in section 2.2.1 of the materials and methods that should give a percentage instead of Kg/colony. Instead, the latter formula would seem to give the results indicated by HB. I suggest the authors to recheck the abbreviations and results.

Pages 7-8 I suggest that authors also bring into the discussion the values that result from the work as they compare themselves to the work of others otherwise the reader must look for them in the work.

-page 8 lines 6-7, 7-9, 10-12, 12-14, 24-27, 31, 31-32, 52-53 bibliographic references are missing. please add them.

-page 8 lines 36-41 Are there any articles to support these assumptions?

-page 9 line 43 “V. destructor (36) and (54). “.  What does article 54 refer to? please add it.

-page 10 conclusions section, line 3 “Varroa” must be written in italics

-supplementary material- please replace “,” with “.” to indicate numbers with decimals

Author Response

First of all, we greatly appreciate the reviewer’s comments.

-page 1 simple summary- line 6, I suggest to change "infestation rates against Varroa" in “Varroa infestation rates”

Done!

-page 1 abstract- lines 5-6 the percentages reported are not those considered in the materials and methods. please check.

Checked and corrected

-page 2 introduction- line 18 (>0.5) unit of measurement is missing. Please add it.

Heritability has not unit 0.5 means 50% of the variability in the trait in a population is due to genetic differences among the population.

-page 2 introduction-lines 27-28, 29-31, 31-32 bibliographic references are missing. please add them.

We added lacking reference!!!

-page 3 materials and methods- 2.2.1 section- why is only the value of the uncapped brooded cells divided by the brood? is it a number that comes out of a division or is it just a way to indicate a unit value? written like that it is not clear.

We reviewed carefully and re-sentence to make it clear

-page 4 materials and methods- 2.2.1 section- lines 1-4 the percentages reported are not those considered in the abstract. please check.

We made the correction!!

-page 4 materials and methods- 2.2.2 section. In the scope of the work the authors talk about Varroa infestation which is the parameter they went to evaluate. Why is it called "bee infestation rates (BIR)"?

We agree, and we did the correction through the text as IR infestation rates as international are reported. We added lacking reference!!!

-page 4 materials and methods- 2.3 section. What tests for normality were applied?

We declared!!!

-page 5 and page 5 figure 1 and 2 Considering that the data are grouped by sampling and altitude they are many so I suggest to apply a correction (example Bonferroni) during statistical analysis.

We did, thanks

-page 6 lines 8-16. “The increase in BIR (average of [….] massive births prior to production.” This whole part should be in the discussion. Please move it.

Done!!!

-page 7 3.4 section- In the text the abbreviation THB is used but it does not seem to correspond to the formula that was provided in section 2.2.1 of the materials and methods that should give a percentage instead of Kg/colony. Instead, the latter formula would seem to give the results indicated by HB. I suggest the authors to recheck the abbreviations and results.

We re-checked and we moved the production values to avoid misinterpretation.

Pages 7-8 I suggest that authors also bring into the discussion the values that result from the work as they compare themselves to the work of others otherwise the reader must look for them in the work.

Done!!!
-page 8 lines 6-7, 7-9, 10-12, 12-14, 24-27, 31, 31-32, 52-53 bibliographic references are missing. please add them.

Added!!!

-page 8 lines 36-41 Are there any articles to support these assumptions?

Done!!

-page 9 line 43 “V. destructor (36) and (54). “.  What does article 54 refer to? please add it.

It refers to the same adult Varroa, we joint both references.

-page 10 conclusions section, line 3 “Varroa” must be written in italics

Done!!!

-supplementary material- please replace “,” with “.” to indicate numbers with decimals

Done!!!

This manuscript is a resubmission of an earlier submission. The following is a list of the peer review reports and author responses from that submission.

Round 1

Reviewer 1 Report

1) The introduction is quite poor and must be extended.

2) Authors should provide a more nuanced interpretation of the results, conclusions are not clear. Errors bars are sometimes huge, so I would appreciate having the source data. Figure must be more detailed in legends.

Author Response

Author's Reply to the Review Report (Reviewer 1)

First of all, the authors would like to give thanks to the reviewers for their suggestion. After we amend the errors, we have seen a huge improvement of the MS.

1) The introduction is quite poor and must be extended.

We have improved our introduction according both reviewers

2) Authors should provide a more nuanced interpretation of the results, conclusions are not clear.

We have reviewed carefully the discussion and conclusions.

3) You said «Accordingly, the reaction of these morphotypes toward mites may be different from the behavior observed in other regions » without reference

We remove this sentence. It is not in the line of the MS

4) In the same way, you said «The importance of the hygienic behavior of honeybee colonies in association with parasite control and the bacterial and mycotic diseases of the brood is well known » without reference.

Included!!

5) “The studies done so far” are not referenced either.

We eliminated all the sentences unrelated with the line of the MS

6) Simple summary : could you explain “ altitudes above 2600 m.a.s.l.” ?

We declare it meter above sea level, Done!

7) Section 2.1 : you said “In this case, three apiaries were excluded”- could you developp it ?

We modified the sentence according your recommendation and the other reviewer.

8) Section 2.3: why were HB data normalized? Please explain what you mean.

We corrected the sentence. THB data were transformed to meet normality creteria.

9) At the end, replace M-W to Mann-Whitney.

Done!

Section 3.1:

10) Errors bars are sometimes huge, so I would appreciate having the source data.

We are sending you the data and we replace the deviation standard bars by standard mean error.

11) Figure must be more detailed in legends.

We added new information to the legends to make them clearer.

Section 3.2: 

12) Figure 2, what is “ab” mean ?

We made the correction ab will be similar to b the third tier.

13) Paragraph 1 : “degree of Africanization of bees” : there is no informations about it, could you provide criteria ? 

We eliminated this paragraph.

14) Paragraph 4 : « « The previous criteria supported the idea of conducting serial analyses of

hygienic behavior throughout the year and determining their means to evaluate honeybee populations in any region », you can not be so affirmative. 

We eliminated this paragraph.

15) « In general, the results of this work suggest that selection is possible both for a higher CH and for higher honey production. », what is « CH » ? You must replace « in general » by an appropriate term.

Done!

16) Conclusions are not cleared. Authors should provide a more nuanced interpretation of the results, not all the time conclusive. For exemple, it is written «According to this result, apiaries located at a lower altitude have a higher level of Africanization and express a higher HB.” ??

As recommended the other reviewer we eliminated all mention of our MS linked with Africanization. Especially in Conclusion.

Reviewer 2 Report

Comments and suggestions:

This is a fairly readable paper, but the wording could be improved in places and the methods and results need clarification.

There is a strong focus on relating the findings to degree of Africanization, including in the discussion, abstract and simple summary, but this is speculative as there is nothing in the paper to quantify Africanization in the sampled bees. I suggest that this aspect should be played down considerably. In paragraph 2 of the introduction, the statement beginning “In the Ecuadoran highlands” is not even referenced. In paragraph 3, “The studies done so far” are not referenced either.

In the simple summary, what is the significance of “ altitudes above 2600 m.a.s.l.”?

The words 2” independently of the Africanization process” should be removed from both here and the abstract.

In the third last line, what does “likewise” mean here?

I suggest the use of "honey bee" throughout rather than “honeybee” as it is more scientifically correct.

In section 2.1, how were the apiaries located? In the point “Honey production per hive (10.2 kg) above the national average [15]” please clarify what the 10.2 kg refers to. You say “In this case, three apiaries were excluded”- does this mean there were 18 to start with?-- if so, please say so.

In section 2.2.1, last sentence, it would be clearer here to refer to the value of THB.

In section 2.2.2, in the formula for BIR the top line would be better as “Number of varroa mites”.

Section 2.3:

why were HB data normalized? Please explain what you mean.

What does “for HB between samples” mean?

The wording “y as a function of x” needs changed throughout this section as correlation treats the two variables equally; it does not treat one as a function of the other.  You could just say “y and x”.

Towards the end, change M-W to Mann-Whitney.

Section  3.1:

Line 1: define in 2.3 how you are presenting descriptive statistics (I presume that 0.7% is standard error but it might not be).

Line 2: are these mean values? Please clarify.

Paragraph 2, “correlating the means of the HB with different altitudes”: do you mean you only used 3 values to get the correlation, one mean per altitude? If so, the statistics needs redone. Usually you use raw values to get correlations. Showing the number of points used when you quote a correlation would make this clearer.

Last sentence in paragraph 2: “According to this result, apiaries located at a lower altitude have a higher level of Africanization and express a higher HB.” This conclusion is not justified by the results. It is speculation. I might explain the result but it is more of a possible explanation for the discussion.

In table 2, what is the typical deviation? It needs more careful definition. Is it the Inter Quartile Range as you also quote the Median?

Just below table 2, sentence beginning “The best hygiene-behavior percentages”, I think this relates to sampling 1 (based on figure 1) but you do not say that. The wording needs made clear.

Section 3.2: 

Below table 3, last line, I think “at the third altitude tier” should be “at the third altitude tier compared to tiers 1 and 2”. Can this be made clear.

Figure 2, what does notation “ab” mean here?

Section 3.4:

Figure 3, it looks unlikely that there is a difference in means for sampling 2: is this correct?

Just below figure 3, again why did you use average HB to get the correlation?

Discussion:

P7, paragraph 1, last sentence, “degree of Africanization of bees”: is there any information on this for the present study?

P8, paragraph beginning “The high prevalence of the mite”, explain the relevance of  breeding programs.

P8, paragraph beginning “Contradictory results”, second last line, change “inferior to” to “lower than”.

P9, second last paragraph, line 4, change “his predecessors” to “its predecessors”.

P10, line 2, define CH.

Second last line, which bees are “these bees”?

Conclusions: I suggest removing “independently of the Africanization process”.

References: these are quite comprehensive but many are in Spanish and not accessible to an international readership. The paper could be of more interest for S. American readers.

Author Response

Author's Reply to the Review Report (Reviewer 2)

First of all, the authors would like to give thanks to the reviewers for their suggestion. After we amend the errors, we have seen a huge improvement of the MS.

This is a fairly readable paper, but the wording could be improved in places and the methods and results need clarification.

Thanks for all your comments and suggestions we believed the MS has improved a lot.

There is a strong focus on relating the findings to degree of Africanization, including in the discussion, abstract and simple summary, but this is speculative as there is nothing in the paper to quantify Africanization in the sampled bees. I suggest that this aspect should be played down considerably.

With your suggestion it was play down.

 In paragraph 2 of the introduction, the statement beginning “In the Ecuadoran highlands” is not even referenced. In paragraph 3, “The studies done so far” are not referenced either.

We eliminated this paragraph.

In the simple summary, what is the significance of “ altitudes above 2600 m.a.s.l.”?

We declare it meter above sea level Done!!!

The words 2” independently of the Africanization process” should be removed from both here and the abstract.

Done!!

In the third last line, what does “likewise” mean here?

We eliminated it.

I suggest the use of "honey bee" throughout rather than “honeybee” as it is more scientifically correct.

Done!

In section 2.1, how were the apiaries located? In the point “Honey production per hive (10.2 kg) above the national average [15]” please clarify what the 10.2 kg refers to.

We moved 10.2 to the end of the sentence referring that 10.2 is the national average

You say “In this case, three apiaries were excluded”- does this mean there were 18 to start with?-- if so, please say so.

We declared it now.

In section 2.2.1, last sentence, it would be clearer here to refer to the value of THB.

Done!

In section 2.2.2, in the formula for BIR the top line would be better as “Number of varroa mites”.

Done!

Section 2.3:

why were HB data normalized? Please explain what you mean.

We corrected the sentence. THB data were transformed to meet normality creteria.

What does “for HB between samples” mean?

The wording “y as a function of x” needs changed throughout this section as correlation treats the two variables equally; it does not treat one as a function of the other.  You could just say “y and x”.

Done!!

Towards the end, change M-W to Mann-Whitney.

Done!

Section  3.1:

Line 1: define in 2.3 how you are presenting descriptive statistics (I presume that 0.7% is standard error but it might not be).

Done!

Line 2: are these mean values? Please clarify.

Done!!

Paragraph 2, “correlating the means of the HB with different altitudes”: do you mean you only used 3 values to get the correlation, one mean per altitude? If so, the statistics needs redone. Usually you use raw values to get correlations. Showing the number of points used when you quote a correlation would make this clearer.

The correlation was done with the straw data. We replace altitude by height to avoid confusing!!!

Last sentence in paragraph 2: “According to this result, apiaries located at a lower altitude have a higher level of Africanization and express a higher HB.” This conclusion is not justified by the results. It is speculation. I might explain the result but it is more of a possible explanation for the discussion.

We agree. We eliminated the sentences.

In table 2, what is the typical deviation? It needs more careful definition. Is it the Inter Quartile Range as you also quote the Median?

We made the corrections. In fact the results are expressed as Mean and SEM.

Just below table 2, sentence beginning “The best hygiene-behavior percentages”, I think this relates to sampling 1 (based on figure 1) but you do not say that. The wording needs made clear.

We eliminated the last two sentences, they were repetitive

Section 3.2: 

Below table 3, last line, I think “at the third altitude tier” should be “at the third altitude tier compared to tiers 1 and 2”. Can this be made clear.

Done!

Figure 2, what does notation “ab” mean here?

We made the correction ab will be similar to b the third tier.

Section 3.4:

Figure 3, it looks unlikely that there is a difference in means for sampling 2: is this correct?

We made the correction

Just below figure 3, again why did you use average HB to get the correlation?

We eliminated this sentence. It didn’t bring new information.

Discussion:

P7, paragraph 1, last sentence, “degree of Africanization of bees”: is there any information on this for the present study?

Agree. We eliminated the sentence.

P8, paragraph beginning “The high prevalence of the mite”, explain the relevance of  breeding programs.

P8, paragraph beginning “Contradictory results”, second last line, change “inferior to” to “lower than”.

Done!

P9, second last paragraph, line 4, change “his predecessors” to “its predecessors”.

Done!!

P10, line 2, define CH.

It is BH, already defined. We changed it.

Second last line, which bees are “these bees”?

It is the bees. We replaced it!

Conclusions: I suggest removing “independently of the Africanization process”.

Done!!

References: these are quite comprehensive but many are in Spanish and not accessible to an international readership. The paper could be of more interest for S. American readers.

We replaced all the Spanish reference that were not available in an English version in the web.
